# O-Arm Navigated Frameless and Fiducial-Less Deep Brain Stimulation

**DOI:** 10.3390/brainsci10100683

**Published:** 2020-09-27

**Authors:** David Krahulík, Martin Nevrlý, Pavel Otruba, Jan Bardoň, Lumír Hrabálek, Daniel Pohlodek, Petr Kaňovský, Jan Valošek

**Affiliations:** 1Department of Neurosurgery, Faculty of Medicine and Dentistry, Palacký University and University Hospital Olomouc, 77900 Olomouc, Czech Republic; lumir.hrabalek@fnol.cz (L.H.); daniel.pohlodek@fnol.cz (D.P.); jan.valosek@fnol.cz (J.V.); 2Department of Neurology, Faculty of Medicine and Dentistry, Palacký University and University Hospital Olomouc, 77900 Olomouc, Czech Republic; martin.nevrly@fnol.cz (M.N.); pavel.otruba@fnol.cz (P.O.); jan.bardon@fnol.cz (J.B.); petr.kanovsky@fnol.cz (P.K.)

**Keywords:** Parkinson’s disease, deep brain stimulation, Nexframe, O-arm

## Abstract

Object: Deep brain stimulation (DBS) is a very useful procedure for the treatment of idiopathic Parkinson’s disease (PD), essential tremor, and dystonia. The authors evaluated the accuracy of the new method used in their center for the placing of DBS electrodes. Electrodes are placed using the intraoperative O-arm™ (Medtronic)-controlled frameless and fiducial-less system, Nexframe™ (Medtronic). Accuracy was evaluated prospectively in eleven consecutive PD patients (22 electrodes). Methods: Eleven adult patients with PD were implanted using the Nexframe system without fiducials and with the intraoperative O-arm (Medtronic) system and StealthStation™ S8 navigation (Medtronic). The implantation of DBS leads was performed using multiple-cell microelectrode recording, and intraoperative test stimulation to determine thresholds for stimulation-induced adverse effects. The accuracy was checked in three different steps: (1) using the intraoperative O-arm image and its fusion with preoperative planning, (2) using multiple-cell microelectrode recording and counting the number of microelectrodes with the signal of the subthalamic nucleus (STN) and finally, (3) total error was calculated according to a postoperative CT control image fused to preoperative planning. Results: The total error of the procedure was 1.79 mm; the radial error and the vector error were 171 mm and 163 mm. Conclusions: Implantation of DBS electrodes using an O-arm navigated frameless and fiducial-less system is a very useful and technically feasible procedure with excellent patient toleration with experienced Nexframe users. The accuracy of the method was confirmed at all three steps, and it is comparable to other published results.

## 1. Introduction

Deep brain stimulation (DBS) is an established technique for modulation of subcortical brain structures in patients with Parkinson’s disease [1,2], essential tremor [3], dystonia [4,5], and other selected movement disorders. Class I evidence supports its use in Parkinson’s disease, in comparison with the best medical treatment [6]. There is a growing tendency to indicate DBS for earlier stages of Parkinson’s disease [7,8]. DBS electrodes have conventionally been placed using frame-based stereotaxis with microelectrode recording (MER) and physiological mapping of target structures. An alternative using a frameless neuronavigation-guided implantation technique with skull-mounted aiming devices (Nexframe™, STarFix™, Clearpoint™) together with bone-implanted fiducial markers is used in some workplaces. A new method using the Nexframe system without fiducials coupled with a perioperative O-arm™ has been developed by Dr. Holloway from Minneapolis, USA. In this technique, the necessary imaging is obtained preoperatively. The O-arm picture is taken at the beginning of the surgical procedure and S8 planning software and navigation is used to register brain targets and planned trajectories. The correct position of electrodes is confirmed by microrecording, macrostimulation, and perioperative O-arm control. Toms et al. published a comparative study of frame-based and fiducial-less procedures [9] and Holewijn et al. also confirmed the sufficient accuracy of the O-arm registration technique used for stereotactic registration [10].

## 2. Methods

Eleven consecutive patients (22 electrodes) were implanted using frameless and fiducial-less technique. All patients were treated for PD and met the Movement Disorder Society Clinical Diagnostic Criteria for Parkinson’s disease [11]. All patients were fully informed about the procedure and the procedure was performed by a single surgeon (D.K.) and neurologists (M.N., P.O.). The accuracy of the procedure was evaluated using the same calculation as in our previous publication of accuracy of frame-based procedures [12] and also with the same method as published by Toms et al., to allow for comparison of our results.

### 2.1. Imaging

Two MRI sets were obtained a few days before the surgery for the PD patients: (1) a volumetric 3D T1 Gd-enhanced gradient echo MRI sequence covering the whole brain in 1 mm axial slices, primarily for trajectory planning and (2) T2 turbo spin echo imaging in 2 mm slices to determine the borders of the subthalamic nucleus (STN). A CT head scan was obtained before surgery for the best fusion with perioperative O-arm imaging.

### 2.2. Surgical Technique

A 3D O-arm (O-arm O2, Medtronic, Inc., Minneapolis, MN, USA) scan was obtained at the beginning of the surgery and fused with preoperative MRI and a CT image in the StealthStation™ S8 (Medtronic, Inc., Minneapolis, MN, USA) stereotactic navigation planning software. Target points for the tips of the electrodes were selected using a combination of direct (visualized) and indirect targeting in Parkinson’s disease. The trajectories were shown using volumetric MRI images on the “navigation” views. Small adjustments to the trajectory were then made to avoid damaging the cortical veins and dural venous lakes (easily identified on Gd-enhanced images) and lateral ventricles. The procedure was performed as a two-stage surgery on the same day. The first stage, implantation of the DBS electrodes, was on an awoken patient, and the second stage proceeded under general anesthesia as the internal pulse generator was implanted.

Using a passive planar blunt probe and active S8 navigation, the burr hole entry point of the predetermined electrode trajectory was then marked on the skin, and a small hole was drilled to mark that point on the skull. After we performed appropriate sterile preparation and draping, linear skin incisions were made, and bur holes centered on the pilot hole were completed. The lead anchoring device (Stimlock, Medtronic) and the Nexframe base were attached to the skull and the navigated O-arm picture was taken and fused. Registration under sterile conditions was performed, achieving target registration error <0.5 mm. The Nexframe tower was then attached and aligned to the corresponding target using S8 navigation software (Medtronic). Target depth was then calculated and set on the microTargeting^TM^ Drive System (FHC) positioning device. The dura was closed with the help of fibrin glue to prevent CSF leak or pneumocephalus.

### 2.3. Intraoperative Microelectrode Registration (MER)

To perform MER in STN-DBS, four MER/macrostimulation needles were placed in an array as central, lateral, anterior, and posterior to delineate the borders of the STN. The starting point for the STN 10 mm above the MRI-based target was set and the microelectrodes were advanced in steps of 500 μm towards the target by an electric microdrive.

### 2.4. Macro-Test Stimulation

After MER, the tip of the microelectrode was retracted. Channels that showed significant multi-unit activity over a length longer than 3 mm were selected for intraoperative test stimulation (60 μs pulse duration; 130 Hz pulse frequency for PD). The complete electrode with the macro-tip was then advanced to be used for macro-test stimulation, and this was performed by an experienced neurologist (M.N., P.O.). After evaluating the selected channels by macro-test stimulation, the one with the largest therapeutic window, i.e., the lowest current threshold for improvement of symptoms and the highest threshold for side effects, was chosen for permanent electrode implantation. Afterwards, the final control 3D O-arm scan was performed after insertion of final lead to confirm its accurate position. The 3D O-arm scan can be used during the surgery several times to confirm accurate position of the microelectrode or the lead. It takes just a few minutes to transfer pictures from the O-arm into the planning station and to fuse the images with CT and MRI. Final control of the position of the electrodes is made by the Suretune™ software (Medtronic, Inc., Minneapolis, MN, USA).

### 2.5. Lead Anchoring and Implantable Pulse Generator Placement

Leads were anchored to the skull with a lead anchoring device (Stimlock, Medtronic). After scalp closure, the surgery continued under general anesthesia and the lead extenders and pulse generators were placed. 

## 3. Results

All 11 patients were implanted using multiple-cell microelectrode recording—four microelectrodes. We recorded the signal from the STN in all patients, which confirmed a good position of the microelectrodes within the STN. The average number of the microelectrodes with good signal was 3 (1 minimum, 4 maximum). After MER, the tip of the microelectrode was retracted. Channels that showed significant multi-unit activity over a length longer than 3 mm were selected for intraoperative test stimulation (60 μs pulse duration; 130 Hz pulse frequency, 1–4 mA for PD).

The next step of confirmation of the proper position of the lead and accuracy of the procedure is obtaining of intraoperative the 3D O-arm scan and its fusion with the preoperative planning. We can control the procedure with this scanning anytime during the surgery (Figure 1 and Figure 2).

The precise location of the electrode within the STN can be evaluated by calculating an error on the preoperative/perioperative MRI/CT fusion images. The entry point AC–PC (anterior commissure–posterior commissure) coordinates (point A) and the target point AC–PC coordinates (point B) of the trajectory are found on the navigation device using preoperative MRI. The target is usually modified intraoperatively according to microrecording and clinical examination by a shift on the trajectory labeled as distance d. Knowing this distance and the AC–PC coordinates of both the starting point and the planned target, it is possible to calculate the AC–PC coordinates of the modified target (point C). The AC–PC coordinates of the actual position of the electrode (point D) are localized by placing a cursor manually at the end of the electrode visible on MRI/CT fusion on the navigation device. Using the equation for calculating the distance of two points in 3D space, it is possible to determine the total error (distance between the modified target and the actual position of the electrode) and by using the equation for the distance of two points on a straight line, the placement errors in the lateral, anteroposterior, and vertical axes are identified. The accuracy measurements are shown in the Table 1. We have compared our results to a previous publication [12] and there were no significant differences between the final results.

We have performed the same calculation as Toms et al. [9] to compare our results. Our results are comparable to the published accuracy in this article (Table 2, Figure 3)

## 4. Discussion

There are generally two methods to perform DBS: one using any stereotactic frame and the other using any frameless system with small fiducials attached to the skull. This new method excludes fiducials and utilizes perioperative O-arm imaging and an online navigation system. None of the systems achieve perfect accuracy and the average error is between 1 and 2 mm. There are several weak points that can lead to inaccuracy with this method, mainly the fusion of MRI, CT, and O-arm, but the newest navigation system has an error of about 1–2 imaging voxels [13]. Urgosik et al. analyzed the accuracy of DBS placement using the Leksell frame according to intraoperative monitoring with very good results and minimum complications [14]. Rohlfing et al., on the other hand, pointed to reduced accuracy of stereotactic frames because of torque introduced by the effect of weight bearing on the frame [15]. Holloway et al. and Krahulik et al. confirmed comparable accuracy of frameless systems to the frame-based systems [12,16]. Recently published articles focusing on the accuracy of O-arm stereotactic registration showed good accuracy of the procedure in the hand of experienced Nexframe users. Our study showed similar results, but the number of patients in our study is smaller. We did not focus this study at surgery time and patients’ toleration of the procedure, but our center has experience with all three procedures (frame-based, frameless Nexframe, fiducial-less Nexframe) and we can say that the procedure is the best tolerated by patients and the operation time is shorter. The disadvantages of this procedure are the radiation dose during O-arm scanning and the longer learning curve to achieve excellent accuracy. All our patients indicated to treat PD, ET, or Dystonia are operated with this procedure at present. 

## 5. Conclusions

The frameless and fiducial-less method using the Nexframe system is an accurate and safe procedure and was well tolerated by our eleven patients. The accuracy of the procedure is comparable to recently published articles and to our previous work focused on a frameless Nexframe approach. It is necessary to obtain a larger dataset to convincingly prove our initial findings and this method should be primary used by experienced Nexframe users.

## Figures and Tables

**Figure 1 brainsci-10-00683-f001:**
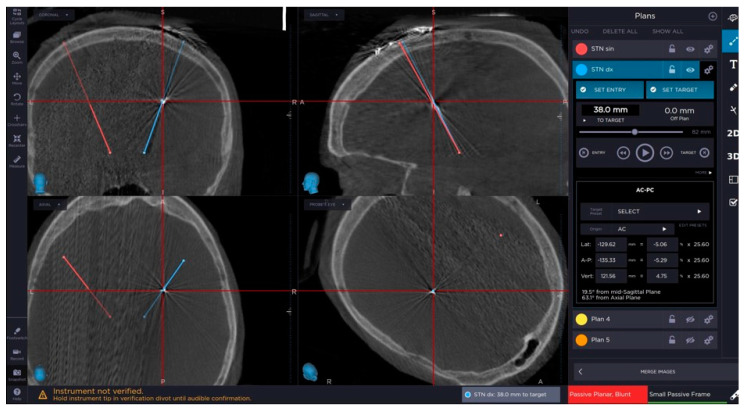
Image from neuronavigational system showing perfect correlation of the actual electrode position and the preoperative planned trajectory.

**Figure 2 brainsci-10-00683-f002:**
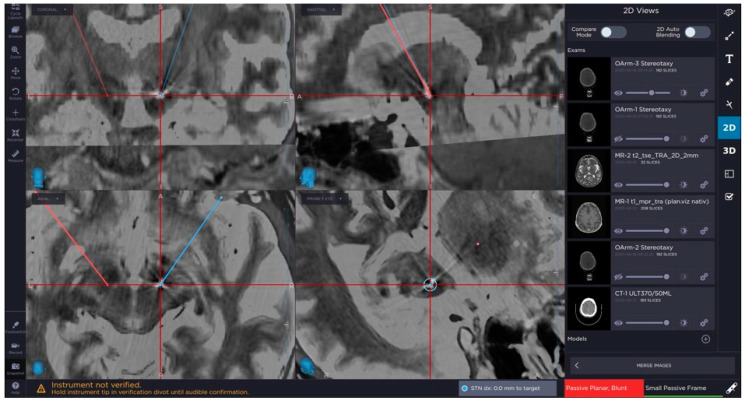
Image from neuronavigational system showing final position of the electrode in the planed target point within the subthalamic nucleus (STN).

**Figure 3 brainsci-10-00683-f003:**
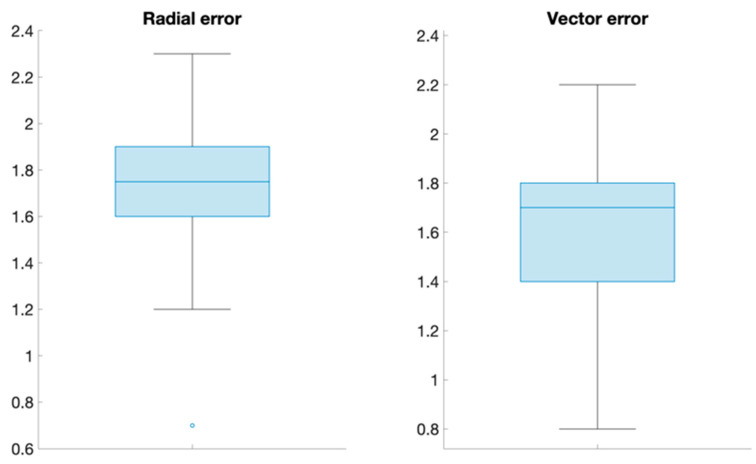
Boxplots showing the radial and vector errors.

**Table 1 brainsci-10-00683-t001:** Accuracy of the procedures FB (fiducial-based) FL (fiducial-less).

Procedure	Total Error (mm)	Lateral Axis	AP axis	Vertical Axis
FL	1.79 ± 0.68	1.10 ± 0.78	1.37 ± 0.87	1.21 ± 0.9
FB	1.64 ± 0.81	1.03 ± 0.79	1.14 ± 0.95	1.05 ± 0.91

**Table 2 brainsci-10-00683-t002:** Final values of the radial and vector errors according to Toms et al. [9].

Error Measure	Total Error (mm)
Radial error	1.71 (0.32)
Vector error	1.63 (0.34)
Radial A-P	1.1 (0.56)
Radial M-L	1.0 (0.34)

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
