# Peer review of "O-Arm Navigated Frameless and Fiducial-Less Deep Brain Stimulation"

_brainsci, 2020, doi:10.3390/brainsci10100683_

Round 1

Reviewer 1 Report

Rewriting by  a native speaker required to fix grammar, spelling errors and style, overall nice paper of moderate novelty, well written otherwise.

Author Response

We have corrected English language and have made various changes according to the second reviewer.

Reviewer 2 Report

This is an interesting topic of interest to neurosurgeons, that seeks to characterize the target accuracy of frameless and fiducialless DBS surgery for Parkinson's Disease. Overall this  article seems like an extension of a previously published article by the same authors in 2019 (which they fail to cite), adding additional data from 6 more patients for a total of 10.  One figure and some of the text is copied from the prior publication.

Strengths:

I believe that significant improvements to the paper would make it interpretable and potentially useful to the field. 10 patients is an  adequate number and the investigators seem experienced using multiple methods in DBS.

Weaknesses:

The Introduction and Discussion should cite a prior study of frameless/fiducialless DBS Targeting: Toms J, Martin S, Sima AP, Chung A, Docef A, Holloway KL. A Comparative Study of Fiducial-Based and Fiducial-Less Registration Utilizing the O-Arm. Stereotact Funct Neurosurg. 2019;97(2):83-93. doi:10.1159/000496810

In the Toms et al study, the authors observed possibly increased radial error (RE) with a fiducialless approach. The current authors need to perform similar calculations to prior articles to allow direct comparison – include the vector distance error etc.. Further, please comment on RE with your approach.

There is PHI / identifying patient information in the screenshots which is inappropriate.

Figures 1 and 2 (picture 1 and picture 2) are redundant, I would replace one of the screenshots with actual analysis. For example, Please show raw data points for each of the 20 electrodes, possibly plotting vector error vs radial error. The formula for simple Euclidean distance should be in methods, not as a figure.

There is no description of Figure 3 abbreviations and what they stand for.

There is no  comparison of the fiducialless  error / accuracy to  fiducial based targeting. The authors have already published data and accuracy from a fiducial based approach in 2017,  it would be relevant to include a summary of this data and statistics comparing the accuracy to the current method  (with citation of course).

The methods for MER recording and actual final values used for test stimulation need to be described in more detail. Which electrode models were used? What were the final amplitudes used and from which contact #s? Provide sufficient detail so someone else can exactly replicate what was done / evaluate possible inconsistencies.

The Discussion is very limited, and no statistics or figures are considered. This needs to be substantially revised to include a discussion  to compare their accuracy values to those previously reported in the literature.Also provide discussion about how one might  decide which surgical approach to use and why. The authors did a much more thorough job in their prior paper from 2017 and 2019.

Author Response

Thank you for your comments. I hope we have made major revision, that will be accepted for publication.

  • we have added recent articles by Toms and Holewijn
  • we have changed pictures to hide identification details of the patient
  • we have added comparison to our previous article to compare to FL method 
  • we have added data for MER , the macrostimulation during surgery  is performed at the tip of the microlelectrode
  • we have made the same calculation according to Toms to compare results
  • we have revised the discussion 

Round 2

Reviewer 2 Report

The authors have responded to all my comments.  I think the paper and analyses are much better. 

Spellcheck needed (e.g. Line 44 'imagining' should be imaging.)

Author Response

I have made spellcheck. Thank you for your comments.